# Effects of different separation methods on the physical and functional properties of extracellular vesicles

**Hyungtaek Jeon, Su-Kyung Kang, Myung-Shin Lee**  *

Department of Microbiology and Immunology, Eulji University School of Medicine, Daejeon, South Korea

* mslee@eulji.ac.kr

**Data Availability Statement:** All relevant data are within the manuscript and its Supporting Information files.

**Funding:** This research was supported by the Mid-Career Researcher Program through the National

## Abstract

Extracellular vesicles (EVs) are small vesicles secreted from cells. They have crucial biological functions in intercellular communications and may even be biomarkers for cancer. The various methods used to isolate EVs from body fluid and cell culture supernatant have been compared in prior studies, which determined that the component yield and physical properties of isolated EVs depend largely on the isolation method used. Several novel and combined methods have been recently developed, which have not yet been compared to the established methods. Therefore, the purpose of this study is to compare the physical and functional differences in EVs isolated using a differential centrifugation method, the precipitation-based Invitrogen kit, the ExoLutE kit, and the Exodisc, of which the latter two were recently developed. We investigated the properties of EVs isolated from non-infected and Kaposi's sarcoma-associated herpesvirus-infected human umbilical vein endothelial cells using each method and determined the yields of DNA, RNA, and proteins using quantitative polymerase chain reaction and bicinchoninic acid assays. Additionally, we determined whether the biological activity of EVs correlated with the quantity or physical properties of the EVs isolated using different methods. We found that Exodisc was the most suitable method for obtaining large quantities of EVs, which might be useful for biomarker investigations, and that the EVs separated using Exodisc exhibited the highest complement activation activity. However, we also found that the functional properties of EVs were best maintained when differential centrifugation was used. Effective isolation is necessary to study EVs as tools for diagnosing cancer and our findings may have relevant implications in the field of oncology by providing researchers with data to assist their selection of a suitable isolation method.

## Introduction

Extracellular vesicles (EVs) are nano-sized membrane vesicles (20–500 nm) of endocytic origin that are secreted by most cells under normal physiological conditions as well as by cells undergoing pathological processes. Based primarily on size and biological origin, EVs can be divided into three main types: (i) apoptotic bodies, which are greater than 800 nm in diameter and are secreted by cells during apoptosis; (ii) microvesicles, which are large membrane-bound vesicles (50–1,000

Research Foundation of Korea (KNRF) funded by the Ministry of Science and ICT (NRF-2019R1A2C2083947, NRF-2017R1A2B4002405) to MSL. NO

**Competing interests:** The authors have declared that no competing interests exist.

nm in diameter) that bud from the plasma membrane; and (iii) exosomes, which are 30–150 nm in diameter and of endocytic origin. In this study, EVs are defined as microvesicles and exosomes.

EVs contain a variety of molecules such as proteins, nucleic acids, and lipids [1–4], which are affected by environmental factors and health conditions [5–7]. EVs exist in body fluids such as cerebrospinal fluid, blood, breast milk, saliva, and urine. Tumor cells secrete more EVs than healthy cells and EVs from tumor cells may have certain tumor-specific markers [8, 9]. Therefore, EVs have been highlighted as potential cancer biomarkers [10, 11]. Furthermore, EV-containing bioactive molecules have an important role in intercellular communication [12, 13]. Many studies have focused on studying the biological activities of EVs and modulating them for therapeutic intervention [14]. For instance, in previous studies, we found that compared with EVs from non-infected cells, EVs from Kaposi's sarcoma-associated herpesvirus (KSHV)-infected human endothelial cells have greater capacity to activate the complement system and that the complement activation activity of these EVs is a biological functional property [15]. Efficient methods of separating and analyzing EVs would provide the potential to better understand their functions, identify their roles in disease, and monitor therapeutic responses.

Despite the importance of EVs as intercellular communicators and potential biomarkers, there are distinct technical challenges in separating and purifying them from biological samples [16, 17]. Proteomics analyses or on-chip assays such as microarray and next-generation sequencing are frequently chosen by many researchers for EV analysis [18–20]. To perform these analyses, a large quantity of well-separated EVs are essential. Differential centrifugation, the classical and most-common method, is generally accepted as the standard [21, 22]. Since differential centrifugation can be applied to most biological fluids and has relatively good reproducibility, this method is most frequently used to isolate EVs from cell culture supernatants or biological fluids [21]. Unfortunately, differential ultracentrifugation cannot process large volumes of samples and its multi-step procedure may compromise the efficiency of processing samples [21]. Furthermore, the requirement of expensive equipment and the length of the processing time is a critical disadvantage for clinical diagnosis applications. In this study, we used differential centrifugation as a control to compare other methods.

Various EV isolation methods have been developed to obtain large, quantities of high-quality EVs. To overcome the drawbacks of ultracentrifugation, precipitation with hydrophilic polymers such as polyethylene glycol (PEG) is also frequently used [23]. Many commercial reagents, such as ExoQuick (System Biosciences) and Total Exosome Isolation reagents (Invitrogen), use PEG for EV isolation. Such precipitation methods do not require expensive equipment and are faster as well as simpler than differential centrifugation. However, the purity of the precipitate can be relatively low, as it may contain non-EV proteins such as albumin, apolipoprotein E, immunoglobulins, and immune complexes [24].

Filtration is the simplest method for separating EVs [25]. Ultrafiltration with membranes that filter out proteins with molecular weights exceeding 100 kDa is frequently used for EV separation [4, 24]. Microfiltration using filters with pore diameters of 0.8, 0.45, 0.1, or 0.02 μm can be used to separate EVs. Larger particles are removed using filters with pore diameters of 0.8 or 0.45 μm, and EVs are then separated using a 0.1 or 0.02 μm filter. Microfilters below 100 nm easily clog during centrifugation, which may cause EV deformation due to the high pressure. Exodisc is a recently developed method designed to separate EVs with low-speed centrifugation to prevent deformation [26].

It may be difficult for a single standard method to serve the purposes of all EV studies; it is likely that different methods are suitable for different EV studies. To investigate the differences in the EV fractions produced by the different methods, EVs were isolated from normal and KSHV-infected human endothelial cells using four different separation methods: 1) the

standard differential ultracentrifugation method; 2) a precipitation-based Total Exosomes Isolation kit by Invitrogen; 3) ExoLutE, a recently developed multistep combined Exosome Isolation Kit involving size-exclusion chromatography by the Rosetta Exosome Company; and 4) a recently developed 20 nm size-selective nanofilter-based isolation method by Exodisc.

## Materials and methods

### Cell culture

Human umbilical cord vein endothelial cells (HUVECs) were purchased from Lonza (Allendale, NJ, USA) and cultured with the endothelial cell growth medium-2 (EGM-2) bullet kit (Lonza, Allendale, NJ, USA) in a humidified atmosphere of 5% $CO_2$ at 37˚C. HUVECs up to passage 6 were used in this study.

### Virus isolation and infection

iSLK BAC16 cells harboring recombinant KSHV BAC16 were used to produce virions [27]. Infectious KSHV BAC16 virions were induced from iSLK BAC16 cells by treatment with doxycycline and sodium butyrate for 3 d. The culture supernatant was collected, filtered through a 0.22 μm filter, and centrifuged at 100 000 × $g$ for 1 h. The pellet was resuspended in phosphate-buffered saline (PBS) and stored at -70˚C as infectious virus particles. HUVECs were infected with KSHV according to the methods used in a previous study [28]. For negative control, mock-infected cells were prepared with the same processes in place of phosphate-buffered saline (PBS) instead of the virus.

### Separation of EVs

The culture supernatant was collected from non-infected and KSHV-infected HUVECs as described previously [15]. Briefly, equal volumes of the culture supernatant were used as the sources of EVs for each of the four different isolation methods. For differential centrifugation, the supernatant was centrifuged at 300 × $g$ for 10 min to remove cellular debris and at 2000 × $g$ for 10 min to remove apoptotic bodies. Subsequently, the supernatant was centrifuged at 10 000 × $g$ for 30 min and then at 100 000 × $g$ for 60 min. The pellet was dissolved with PBS to collect the EVs.

For the other commercial kits and EV separation equipment, we followed the procedures suggested by each manufacturer's instructions. The schematic for the separation process is summarized in Fig 1. For the Invitrogen kit, the supernatant was centrifuged at 2000 × $g$ for 10 min to remove cellular debris and apoptotic bodies. After adding the precipitation solution to the culture supernatant, the mixture was incubated overnight and then centrifuged at 10 000 × $g$ for 60 min. The pellet was dissolved in PBS to collect EVs. For the ExoLutE kit, cellular debris was removed with a 0.45 μm syringe filter and crude EVs were precipitated in the solutions supplied with the kit. The dissolved pellet was processed in a spin-based size exclusion column to separate the EVs. For the Exodisc method, PBS and the culture supernatant were filtered through a 0.45 μm syringe filter. For priming, PBS was added to the filter chamber and centrifuged in a Labspinner centrifuge for 5 min to activate the filter. Then, the clear supernatant was transferred to filter chambers and centrifuged for 5~15 min to separate the EVs for enrichment. Finally, the collected EVs were washed by adding PBS to the filter chambers and centrifuging the solution in the Labspinner. The obtained EVs were used for further analysis.

### Nano-particle Tracking Analysis (NTA)

The number and size distribution of microparticles in the EV preparations were analyzed by the nanoparticle tracking analyzer ZetaView (Particle Metrix GmbH, Meerbusch, Germany).

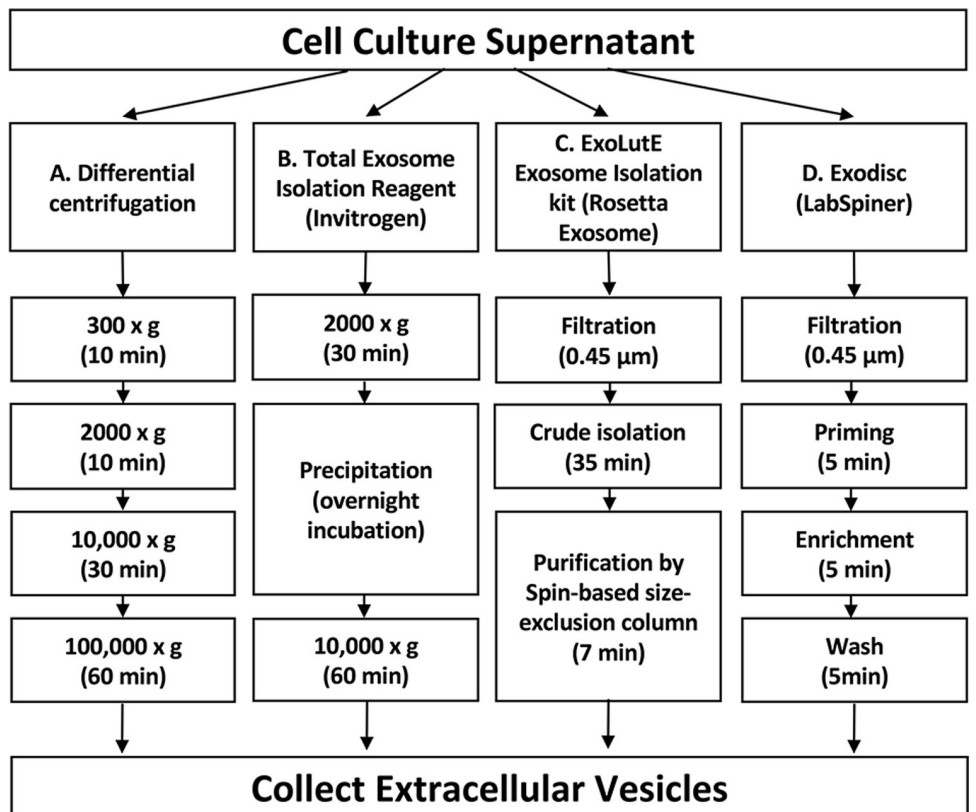

**Fig 1. The schematic summary of the EV separation methods.** The culture supernatant was divided into four parts of equal volumes, each for a different method. (A) Differential centrifugation: the culture supernatant was put through four centrifugation steps to separate EVs. (B) Total Exosome Isolation reagent from Invitrogen: cell debris was removed by centrifugation and EVs were separated by precipitation. (C) ExoLutE Exosome isolation kit from Rosetta Exosome: Cell debris was removed by filtration and EVs were separated by multiple processes, including spin-based size exclusion chromatography. (D) Exodisc from LabSpinner: After filtration to remove cell debris, the supernatant was applied to Exodisc with a 20 nm size-selective nanofilter.

Preparations of EVs were diluted in PBS and passed through 0.8 μm filters before analysis. The analysis parameters were as follows: maximum area: 1000, minimum area: 10, minimum brightness: 25, sensitivity: 75, shutter: 100, and temperature: 25˚C.

## RNA isolation, cDNA synthesis, and quantitative Real-Time Polymerase Chain Reaction (qRT-PCR) analysis

To investigate the quality of the mRNAs, we used an equal amount of mRNA (20 ng) from each preparation for the cDNA synthesis. To analyze the quality of the mRNA, the housekeeping genes GAPDH and β-actin were used as representatives. Subsequently, PCR amplification of different transcripts (GAPDH and β-actin) was performed using specific primer sets (Table 1). Total RNA was isolated using the easy-BLUE total RNA Extraction kit (iNtRON Biotechnology, Daejeon, South Korea) according to the manufacturer's instructions and quantified using Nanodrop-1000 (Thermo Scientific, Waltham, MA, USA). Using a cDNA synthesis kit (Takara, Shiga, Japan), cDNA was synthesized from mRNA. Specific reverse transcription (RT) primers were used for the synthesis of U6 and miR-20a, while random hexamers were used for the synthesis of glyceraldehyde 3-phosphate dehydrogenase (GAPDH) and β-actin. The synthesized cDNA was used as a template for qRT-PCR using the CFX96

**Table 1. List of primers used for PCR.**

| Gene | Sense primer | Antisense primer |
|---|---|---|
| **GAPDH** | GGT ATC GTG GAA GGA CTC | GTA GAG GCA GGG ATG |
| **β-actin** | AGA GCT ACG AGC TGC CTG AC | AGC ACT GTG TTG GCG TAC AG |
| **NADH sub1** | TTC TAA TCG CAA TGG CAT TCC T | AAG GGT TGT AGT AGC CCG TAG |
| **NADH sub5** | TTC ATC CCT GTA GCA TTG TTC G | GTT GGA ATA GGT TGT TAG CGG TA |
| **U6** | CTC GCT TCG GCA CAT ATA CT | ACG CTT CAC GAA TTT GCG TGT C |
| **miR20a** | TAA AGT GCT TAT AGT GCA GGT AG | - |
| **Universal** | - | GTC GTA TCC AGT GCA GGG TCC GAG GT- |

touch real-time PCR detection system (Bio-Rad, Hercules, CA, USA) and TB Green Premix Ex Taq qPCR kit (Takara, Shiga, Japan). The cycling conditions were as follows: 95˚C for 30 s, 40 cycles of 95˚C for 5 s, and 60˚C for 10 s. The specificity of the amplified products was confirmed by analyzing the melting curves. All samples were tested in triplicate and normalized with GAPDH. The primers were synthesized by Genotech (Daejeon, South Korea); their sequences are described in Table 1.

## Isolation of EV DNA

DNA was extracted from isolated EVs using the DNeasy genomic DNA isolation kit (Qiagen, Duesseldorf, Germany). The DNA pellet was resuspended in 100 μL of nuclease-free water. Due to the difficulties in measuring small amounts of DNA with Nanodrop, equal volumes of DNA extracted from the EVs separated by each isolation method were used as a template for qRT-PCR of GAPDH and NADH subunits 1 and 5 with the TB Green Premix Ex Taq qPCR kit (Takara, Shiga, Japan). The cycling conditions were as follows: 95˚C for 30 s, 40 cycles of 95˚C for 5 s, and 60˚C for 10 s. The primer sequences are described in Table 1.

## Protein-based EV quantification and western blot analysis

EVs were lysed in 1x RIPA buffer with the cOmplete[TM] Protease Inhibitor Cocktail (Roche, Basel, Switzerland). The lysate was centrifuged and the supernatants were collected. Protein-based quantification of isolated EVs was performed using the Thermo Scientific protein microBCA assay kit (Rockford, IL, USA). Subsequently, equal volumes of the EVs isolated using different methods were denatured using a 5X sample buffer without dithiothreitol at 95˚C for 10 min and then resolved on 10–12% SDS-acrylamide gel by electrophoresis.

To investigate the composition of known EV markers in EVs separated using each method, normalized quantities (2 μg) of proteins were analyzed by western blotting (S1 Fig). Resolved proteins were transferred onto nitrocellulose membrane from Amersham (GE Healthcare, Cheltenham, GB) and then blocked by incubation in 5% skimmed milk with 0.1% Tween-20 buffer to minimize the non-specific binding of antibodies. Blocked blots were treated with primary antibodies, subsequently washed three times with 1X Tris-buffered saline with 0.1% Tween-20 (TBST) buffer, and then incubated with HRP-conjugated secondary antibody. Unbound antibodies were removed by washing with 1X TBST buffer and were signal-recorded using the West Femto Maximum Sensitivity Substrate Kit under the Bio-Rad ChemiDoc Imager (Hercules, CA, USA).

The antibodies for mouse monoclonal anti-beta-actin (A5316, Sigma-Aldrich, St. Louis, MO, USA), mouse monoclonal CD81 (sc-23962, Santa Cruz Biotechnology, Santa Cruz, CA, USA), mouse monoclonal anti-CD63 (sc-5275, Santa Cruz Biotechnology, Santa Cruz, CA,

USA), rabbit polyclonal anti-HSP70 (ab45133, Abcam, Cambridge, MA, USA), and rabbit polyclonal anti-TSG101 (bs-1365R, Bioss Antibodies Inc., Woburn, MA, USA) were used.

### EV complement activation and C5b-9 cell-ELISA

In our previous study, we found that EVs from KSHV-infected human endothelial cells were able to activate the complement system when they were transferred into non-infected cells [15, 29]. After EVs separated using each method were transferred to human endothelial cells for 24 h, normal human serum was added to supply all complement factors needed to activate the complement system. We evaluated the activation of the complement system by analyzing the deposition of C5b-9 with cell-ELISA and compared the complement system activation potentials among the EVs isolated using the different methods.The cell-ELISA was performed as previously described [30], with modifications. Briefly, 10 000 cells/well were seeded in 96-well culture plates and incubated overnight at 37˚C in 5% $CO_2$. Cells were then cultured in media containing 10% pooled human serum (Innovative Research, Novi, MI, USA) for 1 h to activate the complement system. Plates were washed with PBS and then fixed with 4% paraformaldehyde (PFA) for 15 min. The cells were incubated in blocking buffer (5% skim milk in Tris-buffered saline (TBS)) for 1 h at 37˚C. A rabbit polyclonal C5b-9 antibody (Abcam, Cambridge, MA, USA) diluted in blocking buffer (1:4000) was added to the plate and incubated with the cells for 2 h. The plate was washed three times with TBST for 15 min, and horseradish peroxidase (HRP)-conjugated anti-rabbit IgG (GE Healthcare, Cheltenham, GB) was added. After incubation at room temperature for 1 h, 3,3′,5,5′-tetramethylbenzidine (TMB; KPL, Gaithersburg, MD, USA) was used as a substrate. The absorbance at 450 nm was measured using a microplate reader (Molecular Devices, Silicon Valley, CA, USA).

### Statistical analysis

Each experiment was performed at least three times independently, and representative results are shown. Results are shown as the mean ± the standard deviation (SD). A two-tailed Student's $t$-test and one-way ANOVA were used to assess the significance of the difference between groups. Microsoft Excel (version 16.37) was used for all statistical analyses. Statistical significance at $p$ values of $< 0.05$ and $< 0.01$ is indicated by $*$ and $**$, respectively.

## Results

### EVs separated using different methods demonstrated differences in particle number and size distribution

The number and size distribution of EV particles separated from non-infected and KSHV-infected HUVECs using the different methods are shown in Fig 2. The particle numbers of the isolated EVs differed according to the method used (Fig 2A). Differential centrifugation resulted in the lowest number of particles; the numbers of particles produced by the other methods were about two- to five-fold higher. The EV preparations from each separation method exhibited a range of particle sizes (from 20–500 nm) (Fig 2B). The median size distribution ranged from 120 nm to 140 nm in diameter. Interestingly, the EVs separated using the ExoLutE kit had a larger median size and a broader size distribution than those separated using the other methods.

### Analysis of RNA from EVs separated by different methods

The quantities of RNA from the EVs obtained by the different isolation methods were analyzed (Fig 3A). Interestingly, total RNA quantity was not correlated with EV particle number. While

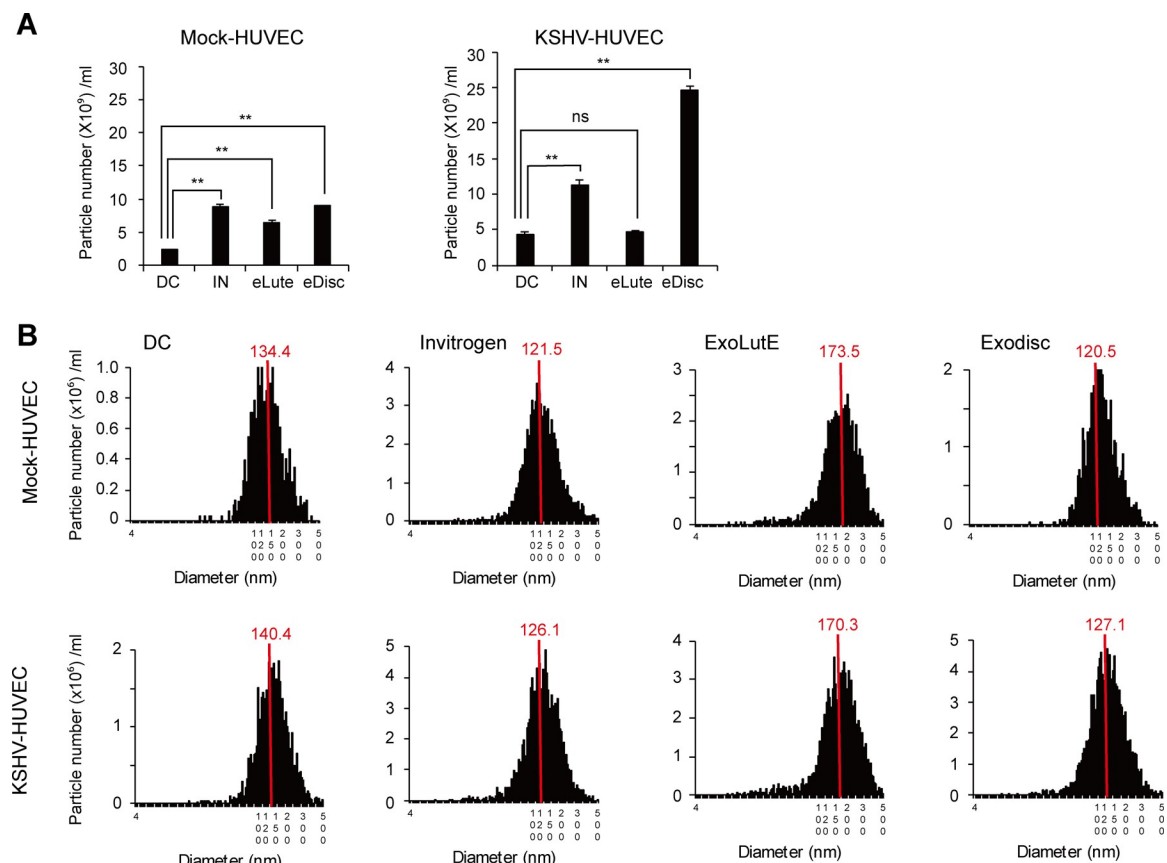

**Fig 2. NTA of EVs separated by different methods.** (A) The number of EV particles separated using each method. DC: EVs separated using differential centrifugation. Invitrogen: EVs separated using the Invitrogen Total Exosome Isolation reagent. eLutE: EVs separated using the ExoLutE exosome isolation kit. eDisc: EVs separated using Exodisc from LabSpinner. Data are shown as the mean ± SD, $n = 3$, *$p < 0.05$, **$p < 0.01$, ns: not significant. (B) Size distribution of EVs separated using each method. The red line indicates the median value of all EV sizes. Mock-HUVEC: mock-infected HUVECs. KSHV-HUVEC: KSHV-infected HUVECs.

the number of EV particles obtained using the Exodisc method was the highest, the quantities of total RNA from EVs separated using differential centrifugation and the Invitrogen kit were larger than that from EVs separated using the Exodisc method. No significant differences were detected between the CT values in the RNAs from different EV preparations, implying that they had qualitatively equal mRNA levels (Fig 3B).

To investigate the quality of non-coding small nuclear RNA (snRNA) and miRNA, total RNA was reverse-transcribed using specific primers for U6 and miR-20a, and the quantity of each type of RNA was analyzed using qRT-PCR (Fig 3C). Intriguingly, the quantities of snRNA and miRNA differed depending on the method. Even though qRT-PCR was conducted with the same quantity of total RNA, the Exodisc and Invitrogen methods produced larger quantities of U6 snRNA and miR-20a than the other methods.

## Analysis of DNA from EVs separated by different methods

The quantities of genomic DNA and mitochondrial DNA analyzed by qRT-PCR using specific primers for GAPDH and NADH subunits 1 and 5, respectively, are shown in Fig 4. qRT-PCR results for GAPDH indicated that the EVs separated using Exodisc had the highest quantity of genomic DNA, followed by those of the Invitrogen kit, differential centrifugation, and the

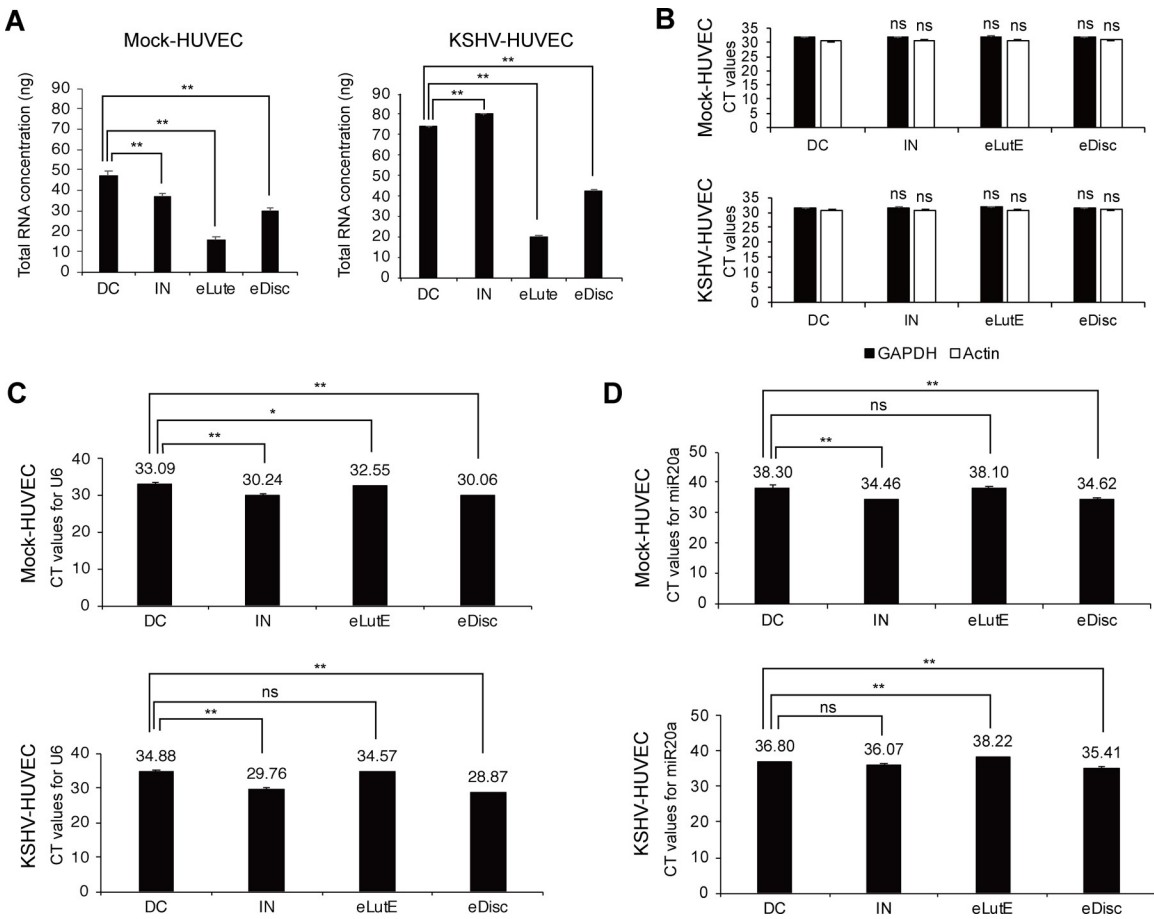

**Fig 3. Analysis of RNA from EVs separated using each method.** (A) Quantification of EV RNA separated using the four different methods. The total quantity of RNA was calculated according to the concentration measured using Nanodrop and is presented based on the EVs separated from 1 mL of culture supernatant. DC: EVs separated using differential centrifugation. IN: EVs separated using the Invitrogen Total Exosome Isolation reagent. eLutE: EVs separated using the ExoLutE exosome isolation kit. eDisc: EVs separated using Exodisc from LabSpinner. Data are shown as the mean ± SD, $n = 3$, $^{**}p < 0.01$. (B) qRT-PCR of GAPDH and β-actin with equal amounts of RNA from the EVs separated by each method. The number above the bar graph indicates the average CT value from the qRT-PCR reaction. CT values from differential centrifugation were used as a control. Data are shown as the mean ± SD, $n = 3$, ns: not significant. (C and D) qRT-PCR for U6 snRNA (C) and miRNA20a (D) with equal amounts of RNA from the EVs separated by each method. The number above the bar graph indicates the average CT value from the RT-qPCR reaction. Mock-HUVEC: mock-infected HUVECs. KSHV-HUVEC: KSHV-infected HUVECs. Data are shown as the mean ± SD, $n = 3$, $^{*}p < 0.05$, $^{**}p < 0.01$, ns: not significant.

ExoLutE kit (Fig 4A). For mitochondrial DNA, the qRT-PCR results for NADH subunits 1 and 5 exhibited a similar pattern to that of the results for GAPDH (Fig 4B). All qRT-PCR reactions presented clear melting curve peaks in the qRT-PCR analyses, indicating that the EVs separated using all four methods contained a notable amount of DNA.

## Analysis of proteins from EVs separated by different methods

To compare the quantities of proteins in the EVs separated by different methods, the proteins in each EV separation were analyzed by bicinchoninic acid assay (BCA) (Fig 5A). Compared to the protein quantity in EVs separated by differential centrifugation, more protein was found in EVs isolated with the Invitrogen kit and less in those isolated with ExoLutE. In the EV fraction separated by Exodisc, a significantly higher amount of protein was found compared to the amount found in EVs separated by the other methods. Although there was a larger amount of

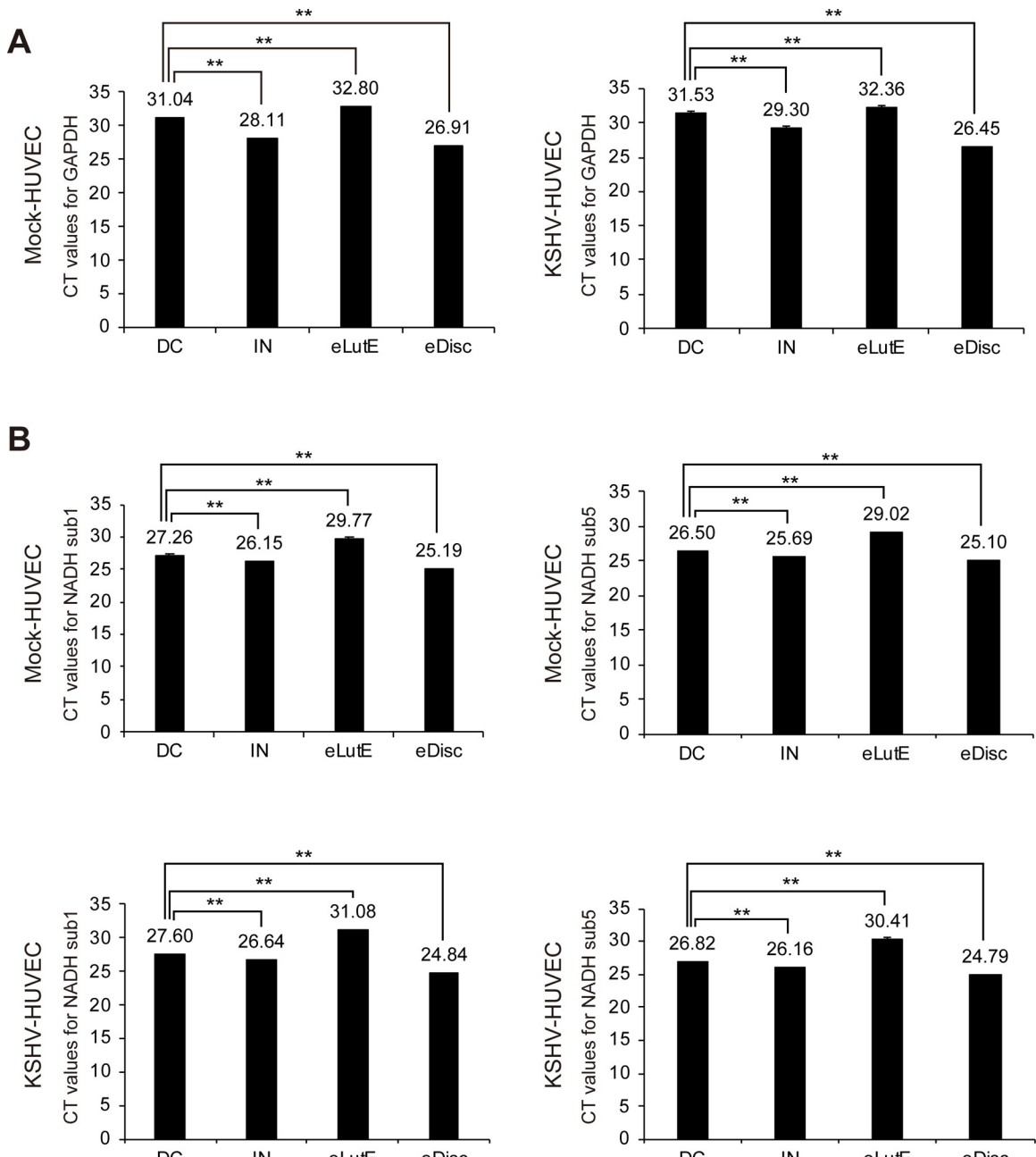

**Fig 4. Analysis of DNA from EVs separated using each method.** EVs were separated from the supernatants of non-infected or KSHV-infected HUVECs using the four different EV separation methods. Equal volumes of DNA extracted from the EVs separated by each method were analyzed with qRT-PCR. qRT-PCR was used to analyze genomic and mitochondrial DNA for GAPDH (A) and NADH subunits 1 and 5 (B), respectively. The number above the bar graph indicates the average CT value from the qRT-PCR reaction. CT values from differential centrifugation were used as a control. DC: EVs separated using differential centrifugation. IN: EVs separated using the Invitrogen Total Exosome Isolation reagent. eLutE: EVs separated using the ExoLutE exosome isolation kit. eDisc: EVs separated using Exodisc from LabSpinner. Mock-HUVEC: mock-infected HUVECs. KSHV-HUVEC: KSHV-infected HUVECs. Data are shown as the mean ± SD, $n = 3$, $^{**}p < 0.01$.

protein found in the EVs from KSHV-infected HUVECs than in EVs from non-infected HUVECs, the pattern of protein quantity extracted from EVs using each method was similar between the EVs from the non-infected and KSHV-infected cells.

The proteins from EVs separated using the Exodisc method were determined to have the highest levels of all EV markers, which was consistent with the results of the BCA (Fig 5B). All EV markers were observed in the EVs separated using differential centrifugation, but only HSP70 was detected in the EVs separated using the Invitrogen kit. In the EVs separated with the ExoLutE kit, none of the EV markers were detected. A previous study showed that EVs from each separation method contain a different composition of EVs marker protein. To investigate the composition of known EVs markers in EVs separated from each method, proteins were analyzed in normalized quantity (2 μg each) by western blot analysis (S1 Fig). EVs from Exodisc showed the highest expression of all markers, and the expression of each marker

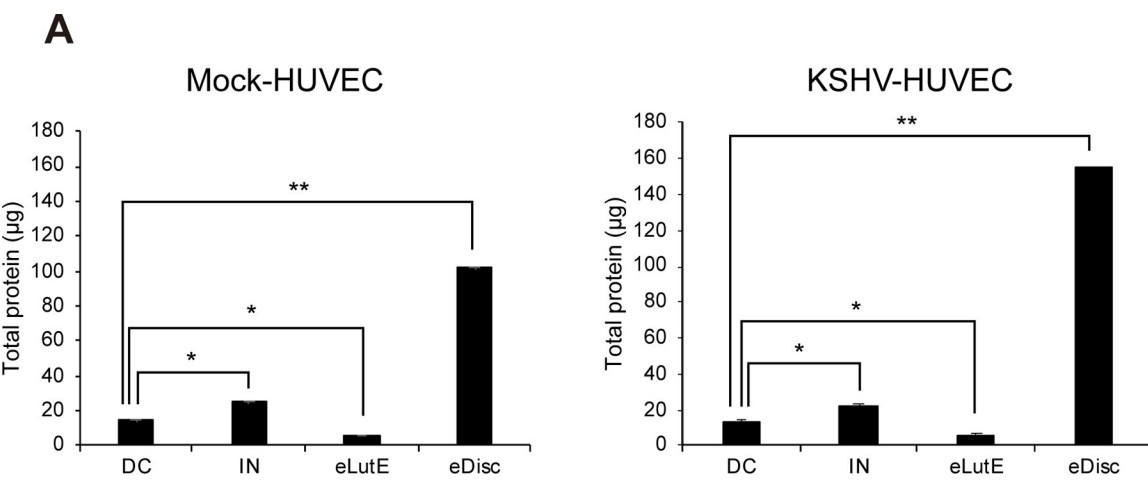

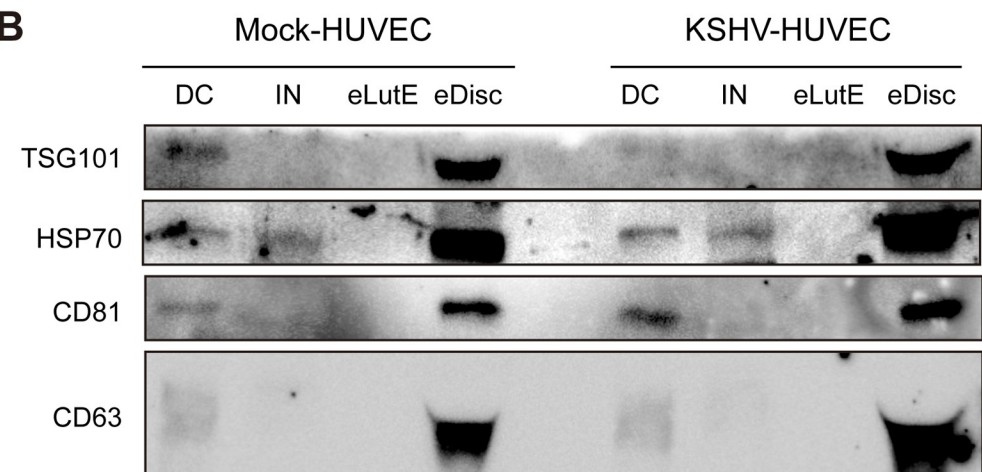

**Fig 5. Analysis of proteins from EVs separated using each method.** (A) Quantification of EV proteins isolated using the four different separation methods. The total quantity of proteins was calculated using BCA and is presented based on the EVs separated from 1 mL of culture supernatant. DC: EVs separated using differential centrifugation. IN: EVs separated using the Invitrogen Total Exosome Isolation reagent. eLutE: EVs separated using the ExoLutE exosome isolation kit. eDisc: EVs separated using Exodisc from LabSpinner. Data are shown as the mean ± SD, $n = 3$, $^*p < 0.05$, $^{**}p < 0.01$. (B) Western blot analysis of EV markers. Equal volumes of EVs separated by each method were loaded onto the gel, and EV markers were analyzed by western blotting. Mock-HUVEC: mock-infected HUVECs. KSHV-HUVEC: KSHV-infected HUVECs.

was different according to the method, indicating that the composition of EVs extracted by each separation method may be different.

## Complement activation of EVs separated by different methods

In the cells treated with EVs from non-infected cells, complement activation was not observed as expected. The EVs separated from KSHV-infected cells using the Exodisc method showed the highest amount of deposition. Interestingly, the EVs separated using differential centrifugation had a slightly lower amount of deposition compared to those separated using the Exodisc method. The EVs separated using the Invitrogen kit and the Exolute kit also activated the complement system, although the amounts of deposited C5b-9 were lower than those of the EVs separated using differential centrifugation and the Exodisc method (Fig 6A and 6B).

## Discussion

We found that the four different separation methods yielded EVs with different physical and functional properties. The characteristics of the separated EV fractions appeared to vary; the number and size distribution of the EV particles isolated using different methods exhibited some differences.

For instance, total RNA from the samples separated using the different methods was not consistent with the number of EV particles in each sample and miR20a from the same amount of total RNA was significantly higher in EVs separated using the Exodisc method. Since the quantity of DNA in the EVs was too small to measure with a spectrophotometer, qRT-PCR was used for analysis. The qRT-PCR results shown in Fig 4 demonstrated significant quantitative differences in genomic and mitochondrial DNA. EVs isolated with Exodisc had the highest quantity of DNA. However, relative to the EV particle number, the purity of the EVs isolated using the Invitrogen kit was higher.

In a previous study comparing the Invitrogen kit with differential centrifugation, the Invitrogen kit produced a higher yield of total protein and RNA than that produced by differential centrifugation due to the presence of non-EV proteins; the Invitrogen preparation also showed a broader size distribution [31]. Our results were consistent with this prior finding and also showed that Exodisc was more suitable than the other methods for obtaining larger quantities of EV proteins.

Notably, in our study, the highest yield of EV particles was obtained using the Exodisc method, possibly due to its ability to concentrate all nanoparticles with a diameter over 20 nm. Therefore, Exodisc might be the best method to obtain large quantities of EVs, as the particle number obtained using this method was significantly higher than those obtained using the other methods. This method would be very useful for identifying biomarkers in EVs because a large amount of EVs were separated from a comparatively small biological sample. The quantity of protein and DNA in the isolated EVs correlated positively with the particle number obtained by each method. These findings supported the results of a previous study showing that EVs separated by different methods contain different compositions of EV marker proteins [32].

In contrast to the other methods, the ExoLutE kit used in this study uses a combination of multiple methods, including precipitation and size-exclusion chromatography. Size-exclusion chromatography can produce an EV fraction with a higher purity by generating samples free of non-EV proteins and lipoproteins [32]. However, this method requires pretreatment and concentration of EV samples by ultracentrifugation or ultrafiltration. Furthermore, a relatively low yield is a disadvantage of this multi-step separation process; although the purity achieved was high, the yield of the ExoLutE kit was the lowest among the four methods.

**A**

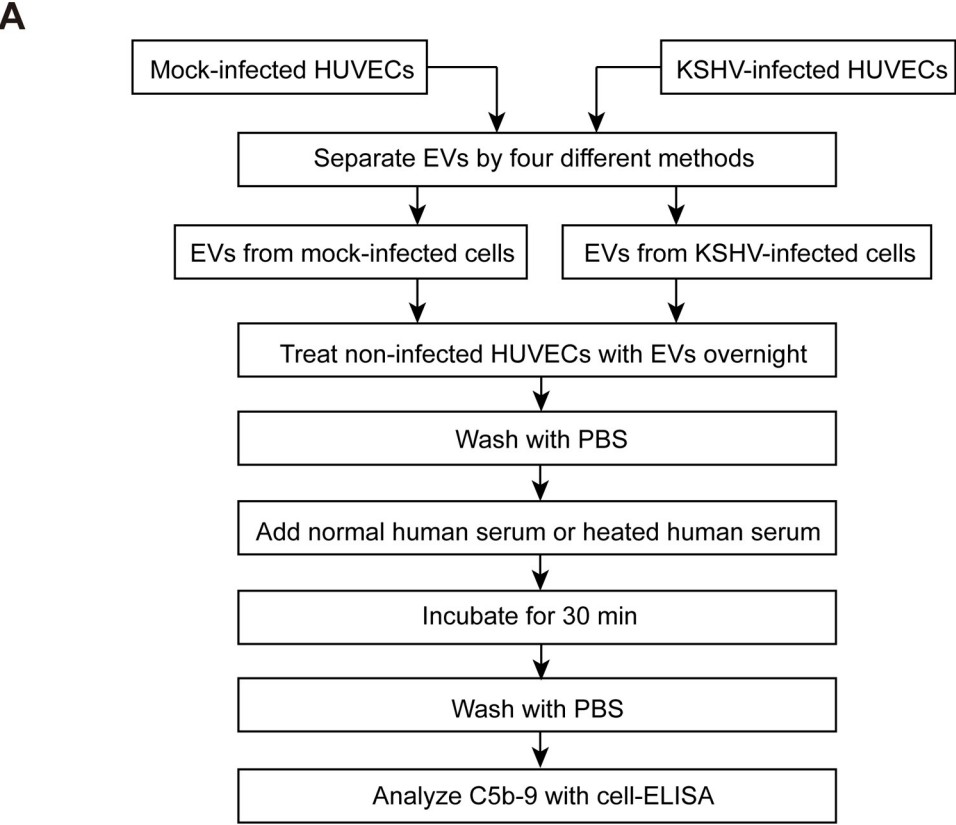

**B**

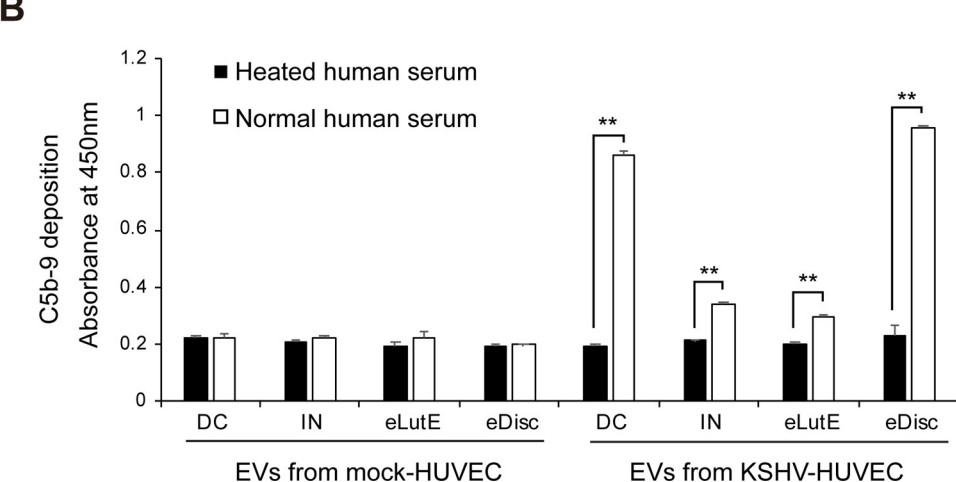

**Fig 6. Activation of the complement system by EVs isolated from KSHV-infected cells using different methods.**
(A) Schematic summary of the experimental process. Equal volumes of EVs separated by the four different methods from KSHV-infected HUVECs were applied to non-infected HUVECs with heated or normal human serum. The deposition of C5b-9 on the cells was analyzed by cell-ELISA. (B) The results of the cell-ELISA for C5b-9 in the non-infected HUVECs treated with separated EVs by various methods. DC: EVs separated using differential centrifugation. IN: EVs separated using the Invitrogen Total Exosome Isolation reagent. eLutE: EVs separated using the ExoLutE exosome isolation kit. eDisc: EVs separated using Exodisc of LabSpinner. Data are shown as the mean ± SD, $n = 3$, $^{**}p < 0.01$.

In terms of biological function, the EVs separated using Exodisc showed the highest level of complement activation. In a previous study using differential centrifugation, KSHV-infected HUVECs exhibited EV production and complement system activation potential that increased relative to those of the non-infected HUVECs [15]. Surprisingly, in our study, EVs isolated using differential centrifugation had similar complement activity as those isolated with Exodisc, even though the number of EV particles isolated with differential centrifugation was much lower than that of EV particles isolated with Exodisc. This result indicated that the purity of the EV sample isolated using differential centrifugation would be higher than that of the same isolated using Exodisc.

A limitation of this study was that although the samples tested with each method were taken from the same source and should have had similar qualities, each sample had its own unique characteristics. EVs are a heterogenous population containing exosomes, microvesicles, and apoptotic bodies; even among EVs of the same size, individual EVs may have differing contents of DNA, RNA, and protein. Therefore, it may be impossible to isolate a homogenous population of EVs. Each EV separation method has a different theoretical background, and our results showed that EVs isolated using different methods had different physical and functional properties. However, it can be difficult to determine to what degree the differences in isolated EV samples are due to the effects of EV population heterogeneity and to what degree they can be attributed to the choice of isolation method. Additionally, while our study showed that the different methods produced separated EV fractions with significantly different physical and functional properties, further studies that include miRNA profiling, protein profiling, and more functional assays of EVs isolated with each method are necessary.

In conclusion, we provided novel data that showed how the recently developed ExoLutE and Exodisc methods compared to established methods. We demonstrated that the newly developed Exodisc method yielded the largest quantity of EVs; this indicated that Exodisc might be especially useful when searching for biomarkers. Selecting an appropriate separation method for a given purpose may be critical not only for producing useful results but also for reproducing experiments; our findings may help researchers determine which methods best suit their needs.

## Supporting information

**S1 Fig. Western blot analysis of markers from EVs normalized for quantification.** Two micrograms of protein lysate from each sample of EVs separated using the four different methods were loaded to the gel. The EV markers Tsg101, CD63, CD81, and HSP70 were analyzed by Western blotting.
(TIF)

**S2 Fig. Western blot uncut image of Fig 5B.**
(TIF)

**S1 Dataset.**
(XLSX)

## Acknowledgments

We thank the members of Lee's laboratory for technical assistance and helpful discussions.

## Author Contributions

**Conceptualization:** Myung-Shin Lee.

**Data curation:** Myung-Shin Lee.

**Formal analysis:** Hyungtaek Jeon, Su-Kyung Kang.

**Funding acquisition:** Myung-Shin Lee.

**Investigation:** Myung-Shin Lee.

**Methodology:** Hyungtaek Jeon, Su-Kyung Kang.

**Resources:** Hyungtaek Jeon, Su-Kyung Kang.

**Supervision:** Myung-Shin Lee.

**Validation:** Hyungtaek Jeon, Su-Kyung Kang.

**Visualization:** Su-Kyung Kang.

**Writing – original draft:** Hyungtaek Jeon.

**Writing – review & editing:** Myung-Shin Lee.

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
