## [Decision Letter · Decision Letter 0]

15 May 2020

PONE-D-20-09529

Effects of different separation methods on the physical and functional properties of extracellular vesicles

PLOS ONE

Dr. Lee,

Thank you for submitting your manuscript to PLOS ONE. After careful consideration, we feel that it has merit but does not fully meet PLOS ONE’s publication criteria as it currently stands. Therefore, we invite you to submit a revised version of the manuscript that addresses the points raised during the review process.

I require that all controls associated with western blots, EV isolation, and other techniques used in the paper must be provided in the revised submission. Additional revisions to the write up is recommended, especially the rationale for the study, and its objective. Statistical analysis needs to be performed throughout the manuscript, replicates (biological vs. technical), all details of the methods used must be included. I would encourage the manuscript be proofread by professional services before resubmission. Finally, data that helps determine whether the preferred EV isolation method is suitable for downstream analysis will greatly strengthen the manuscript findings. However, this data request is optional. 

We would appreciate receiving your revised manuscript by August 15, 2020. If you need more time to make these revisions, please do not hesitate to contact us. To enhance the reproducibility of your results, we recommend that if applicable you deposit your laboratory protocols in protocols.io, where a protocol can be assigned its own identifier (DOI) such that it can be cited independently in the future. For instructions see: http://journals.plos.org/plosone/s/submission-guidelines#loc-laboratory-protocols

We look forward to receiving your revised manuscript.

Sincerely,

Ramani Ramchandran

Academic Editor

PLOS ONE

Journal Requirements:

Reviewers' comments:

Reviewer's Responses to Questions

**Comments to the Author**

1. Is the manuscript technically sound, and do the data support the conclusions?

Reviewer #1: Partly

Reviewer #2: Partly

2. Has the statistical analysis been performed appropriately and rigorously? 

Reviewer #1: No

Reviewer #2: I Don't Know

3. Have the authors made all data underlying the findings in their manuscript fully available?

Reviewer #1: Yes

Reviewer #2: No

4. Is the manuscript presented in an intelligible fashion and written in standard English?

Reviewer #1: Yes

Reviewer #2: No

5. Review Comments to the Author

Reviewer #1: Rationale for "Comments to the Author":

1. Is the manuscript technically sound, and do the data support the conclusions?

Please include or explain the lack of HUVECs treated with a vehicle control for the experiment associated with figure 6

Please alter lines 355-357 of page 16, as you are not determining which isolation method is "most suitable," but rather just documenting the physical and functional effects of each

2. Has the statistical analysis been performed appropriately and rigorously?

What software and model is used? Also, t-tests are inappropriate for the number of experimental groups. Look towards one/two-way ANOVAs

3. Have the authors made all data underlying the findings in their manuscript fully available?

The data statement is included

4. Is the manuscript presented in an intelligible fashion and written in standard English?

The manuscript is easy to follow

Additional Comments:

Please include citations (line 57-58 on page 3; lines 68-71 and 77 of page 4, and lines 381-383 of page 17) for these statements/ claims.

Please explain the types of EVs and which you will be isolating in your introduction (exosomes, micorvesicles, and/or apoptotic bodies)

Please move your list of antiboides located under "cell culture and antibodies" to the section titled "protein-based centrifugation and western blot analysis"

Please state the passage number used of HUVECs

Please move the latter portion of lines 232-234 to the discussion section (speculation belongs in the discussion)

Please briefly summarize the findings found in reference 29 (lines 357-358 on page 16)

Please also include in the discussion that it is difficult to make decisions without: miRNA profiling, protein profiling, and more functional assays

Reviewer #2: Summary

The goal of this research was to find the method that will best facilitate identifying EV biomarkers and thus identifying the method that is the most suitable for “follow-up studies involving EVs.” The authors isolated EVs by new and traditional approaches including differential ultracentrifugation, PEG precipitation with two new EV isolation technologies Exodisc and Exolute. For each isolation method then compared select DNA, RNA, and protein cargos as well as their ability to induce compliment activation. They found that each method distinct outcomes, isolating different amounts of particles with distinctive mean sizes, with different levels of select protein, RNA, and DNA cargos. They went on to show that the EV isolated from herpes-infected cells though the four methods induce different levels of compliment activation in naïve cells.

Major Concerns

As an EV biologist I can appreciate the findings in this manuscript. The Exodisc, and Exolute purification methods were news to me. It’s nice that they were able to directly compare these novel methods with two widely used purification methods. However, I was curious why they did not also compare immunoprecipitation as this is increasingly becoming the purification method of choice. The key finding of interest for me was that the Exodisc recovers way more EVs than other methods. Their results clearly show that the particles purified by the Exodisc contain traditional protein cargos of EVs and that these contain classical EV protein markers.

While the authors conducted rigorous experiments that each showed extremely little variation between biological replicates. As it stands, they are only quantifying a few select RNA, DNA transcripts. It would be nice to have seen a broader characterization of the protein and RNA. In the background they describe their motivation is to identify a purification method that empowers RNAseq and LC-MS-MS analysis of EVs. It would have been nice ideal to have included some -omic data. Later in the discussion they establish their aim to determine “which method is most suitable for a specific research, specifically, we investigated which method would be most suitable for identifying biomarkers in EVs.” Later they change and say “It may be difficult for a single standard method to serve the purposes of all EV studies. We hypothesized that different methods are suitable for different EV study purposes.” And concluded with “In this study, physical and functional properties were analyzed and compared in EVs obtained using the different separation methods to demonstrate which methods would be suitable for follow-up studies involving EVs.” This is pretty vague, but even given this broad goal it’s not clear how these experiments are anything but the first step towards that. While the experiments are well done I feel that while these results are useful and necessary for the research group to develop their experimental methodology I don’t think that the novelty and impact of them is sufficient for publishing in PlosOne.

Minor Concerns

The methods for EV purification reference other publications. It would be nice to have them at least described briefly. This seems especially important since the focus of the paper is on addressing the results of these methods.

The figures are difficult to follow without figure legends.

For each experiment it would be nice to indicate the number of biological replicates.

Nanoparticle tracking analysis of EVs can be notoriously prone to artifacts and strongly dependent on setting conditions. Therefore, it would have been nice to provide a complimentary characterization of EV abundance, such as also show electron micrographs of EVs.

Line 163 “random RT primers were used for synthesis of 164 glyceraldehyde 3-phosphate dehydrogenase (GAPDH) and β-actin.” What do you mean by random? Primers of random sequence would not specifically amplify GAPDH or B-actin.

Also, the authors should make a case for why they chose to look at these transcripts.

They never state the number of particles they isolated with the different methods and the lo-res figures make it impossible to make out the exponential on the Y-axis.

There is little discussion on the significant differences they found in DNA and RNA cargo abundance. It would be nice to hear how finding differences in these nucleotide cargos furthers the overall goal of the research.

It would be nice to see Western blots also conducted on samples normalized for protein. As it is the Western blot results in figure 5B could be simply be due to the differential abundance of the starting samples.

The authors do not make clear if their biological activity of inducing complementation is definitively associated specifically with EVs or there may also be other secreted factors that induce it. This knowledge is critical to make any kind of determination of EV bioactivity. Also it is curious that the ultracentrifuged EVs, with many less particles and lower EV protein makers nevertheless had similar activity to the much more abundant Exodisc sample. This would seem to show that the ultracentrifuged EVs was much purer in terms of activity. Maybe this could be due to a ceiling effect and which they could address with a dilution series.

Grammar issues throughout.

6. PLOS authors have the option to publish the peer review history of their article (what does this mean?). If published, this will include your full peer review and any attached files.

Reviewer #1: No

Reviewer #2: No

---

## [Author Response · Author response to Decision Letter 0]

1 Jun 2020

CC: rramchan@mcw.edu

PONE-D-20-09529

Effects of different separation methods on the physical and functional properties of extracellular vesicles

PLOS ONE

Dr. Lee,

Thank you for submitting your manuscript to PLOS ONE. After careful consideration, we feel that it has merit but does not fully meet PLOS ONE’s publication criteria as it currently stands. Therefore, we invite you to submit a revised version of the manuscript that addresses the points raised during the review process.

I require that all controls associated with western blots, EV isolation, and other techniques used in the paper must be provided in the revised submission. Additional revisions to the write up is recommended, especially the rationale for the study, and its objective. Statistical analysis needs to be performed throughout the manuscript, replicates (biological vs. technical), all details of the methods used must be included. I would encourage the manuscript be proofread by professional services before resubmission. Finally, data that helps determine whether the preferred EV isolation method is suitable for downstream analysis will greatly strengthen the manuscript findings. However, this data request is optional. 

We would appreciate receiving your revised manuscript by August 15, 2020. If you need more time to make these revisions, please do not hesitate to contact us. To enhance the reproducibility of your results, we recommend that if applicable you deposit your laboratory protocols in protocols.io, where a protocol can be assigned its own identifier (DOI) such that it can be cited independently in the future. For instructions see: http://journals.plos.org/plosone/s/submission-guidelines#loc-laboratory-protocols

 • A rebuttal letter that responds to each point raised by the academic editor and reviewer(s). This letter should be uploaded as separate file and labeled 'Response to Reviewers'. 

 • A marked-up copy of your manuscript that highlights changes made to the original version. This file should be uploaded as separate file and labeled 'Revised Manuscript with Track Changes'. 

 • An unmarked version of your revised paper without tracked changes. This file should be uploaded as separate file and labeled 'Manuscript'. 

We look forward to receiving your revised manuscript.

Sincerely,

Ramani Ramchandran

Academic Editor

PLOS ONE

Authors’ response:

We thank the Academic Editor for the consideration and the favorable comments. We have revised our manuscript according to the comments of the Editor and the reviewers.

Controls associated with western blots, EV isolation, and other techniques

In this study, differential centrifugation was used as a control method; the results of the other methods were compared with the results of the differential centrifugation. In the KSHV infection experiments, we used mock-infected cells as a control and prepared them using the exact same process used to prepare the KSHV-infected cells. We apologize for the confusion caused by not clearly describing the mock-infected HUVECs; we have changed “HUVECs” in all figures to “mock-HUVECs” and corrected the manuscript text accordingly. 

The objective and rationale for the study

We have revised the abstract and discussion as follows:

Abstract line 23-26 on page 2:

“Therefore, the purpose of this study is to compare the physical and functional differences in EVs isolated using a differential centrifugation method, the precipitation-based Invitrogen kit, the ExoLutE kit, and the Exodisc, of which the latter two were recently developed.”

Discussion line 462-468 on page 21:

“In conclusion, we provided novel data that showed how the recently developed ExoLutE and Exodisc methods compared to established methods. We demonstrated that the newly developed Exodisc method yielded the largest quantity of EVs; this indicated that Exodisc might be especially useful when searching for biomarkers. Selecting an appropriate separation method for a given purpose may be critical not only for producing useful results but also for reproducing experiments; our findings may help researchers determine which methods best suit their needs.” 

Statistical analysis throughout the manuscript, replicates (biological vs. technical):

Statistical analysis information has been uploaded to the S1 Data set of Supplementary Information file.

Methods details:

Details of the methods have been extensively revised.

Line 138-150 on page 7:

“For the Invitrogen kit, the supernatant was centrifuged at 2000 × g for 10 min to remove cellular debris and apoptotic bodies. After adding precipitation solution to the culture supernatant, the mixture was incubated overnight and then centrifuged at 10 000 × g for 60 min. The pellet was dissolved in PBS to collect EVs. For the ExoLutE kit, cellular debris was removed with a 0.45 μm syringe filter, and crude EVs were precipitated with solutions supplied by the ExoLutE kit. The dissolved pellet was put into a spin-based size exclusion column to separate the EVs. For the Exodisc method, PBS and the culture supernatant were prepared by filtering with a 0.45 μm syringe filter. For priming, PBS was added to the filter chamber and centrifuged in a Labspinner centrifuge for 5 min to activate the filter. Then, for enrichment, the clear supernatant was transferred to filter chambers and centrifuged for ~5–15 min to separate the EVs. Finally, for washing, the collected EVs were washed by adding PBS to the filter chambers and centrifuging the solution in Labspinner. The obtained EVs were used for further analysis.” 

Proofreading by a professional service before submission:

The manuscript has been edited by a professional English editing company. Please see the editing certificate.

-Omics data

We thank the Editor for the excellent suggestions and agree that -omics data would be ideal.

Unlike studies using cancer cells, it is challenging to obtain a large amount of EVs from KSHV-infected primary cells. Since it was difficult to obtain enough EV proteins for proteomics analysis using known EV separation methods, several newly developed methods were sought to overcome this limitation. In fact, we did conduct a proteomics study with EVs isolated from KSHV-infected cells using the Exodisc method, but it was difficult to obtain enough isolated EVs using the other methods for LC-MS-MS analysis. It would be ideal to present the -omics data of EVs extracted by different methods, but we are very sorry to report that we are unable to apply them this time due to our technical and financial limitations.

Journal Requirements:

Authors’ response: 

We have carefully edited the manuscript to meet PLOS ONE’s style requirements according to the templates. 

Authors’ response: 

We have uploaded the minimal data set as a Supporting Information file.

Authors’ response: 

We have uploaded the uncut blot images as S2 Fig.

Reviewers' comments:

Reviewer's Responses to Questions

Comments to the Author

1. Is the manuscript technically sound, and do the data support the conclusions?

Reviewer #1: Partly

Reviewer #2: Partly

2. Has the statistical analysis been performed appropriately and rigorously?

Reviewer #1: No

Reviewer #2: I Don't Know

3. Have the authors made all data underlying the findings in their manuscript fully available?

Reviewer #1: Yes

Reviewer #2: No

4. Is the manuscript presented in an intelligible fashion and written in standard English?

Reviewer #1: Yes

Reviewer #2: No

5. Review Comments to the Author

Reviewer #1: Rationale for "Comments to the Author":

1. Is the manuscript technically sound, and do the data support the conclusions?

Please include or explain the lack of HUVECs treated with a vehicle control for the experiment associated with figure 6

Authors’ response:

We thank the Reviewer for the valuable suggestion. We apologize for the confusion regarding the control. As a control, we used EVs from mock-infected HUVECs (HUVECs treated with a vehicle control; indicated as just “EVs from mock-HUVEC” in Figure 6). 

To activate the complement system, we used normal human serum. As a control, we used serum inactivated by heating at 56 °C for 30min. More detailed experimental procedures can be found in our previous papers (Jeon, H., Lee, J.S., Yoo, S. & Lee, M.S. Quantification of complement system activation by measuring C5b-9 cell surface deposition using a cell-ELISA technique. J. Immunol. Methods 415, 57–62 (2014), and Jeon, H. et al. Extracellular vesicles from KSHV-infected endothelial cells activate the complement system. Oncotarget (2017). doi:10.18632/oncotarget.21668). 

To further assist with the reader’s understanding, a schematic diagram of the experimental process has been added to Figure 6 (Fig 6A). 

Please alter lines 355-357 of page 16, as you are not determining which isolation method is "most suitable," but rather just documenting the physical and functional effects of each

Authors’ response:

We completely agree with the reviewer’s excellent observation and have changed the description of the study’s contribution (now at line 462-468):

“…we provided novel data that showed how the recently developed ExoLutE and Exodisc methods compared to established methods. We demonstrated that the newly developed Exodisc method yielded the largest quantity of EVs than other methods; this indicated that Exodisc might be especially useful when searching for biomarkers.” 

2. Has the statistical analysis been performed appropriately and rigorously?

What software and model is used? Also, t-tests are inappropriate for the number of experimental groups. Look towards one/two-way ANOVAs

Authors’ response: 

We used Microsoft Excel for all statistical analyses. We agree that t-tests are not appropriate for the number of experimental groups. However, in this study, we tried to show the differences between EVs separated using newly developed methods and those separated using the standard method. To accomplish this purpose, we used EVs separated using differential centrifugation as a control. In this setting, the t-test is looking more useful than ANOVA to compare each method with differential centrifugation. Nevertheless, we have added the statistical results of the multiple groups using ANOVA to the data set in the Supporting Information file.

3. Have the authors made all data underlying the findings in their manuscript fully available?

The data statement is included

4. Is the manuscript presented in an intelligible fashion and written in standard English?

The manuscript is easy to follow

Additional Comments:

Please include citations (line 57-58 on page 3; lines 68-71 and 77 of page 4, and lines 381-383 of page 17) for these statements/ claims.

Authors’ response: 

We apologize for these mistakes. We have added the citations as the reviewer suggested.

Line 57-58 on page 3 � line 71-72 on page 4, references 21, 22

Lines 68-71 and 77 of page 4 � line 82-83 on page 4, reference 23

Lines 381-383 of page 17 � line 429-431 on page 20, reference 32

Please explain the types of EVs and which you will be isolating in your introduction (exosomes, micorvesicles, and/or apoptotic bodies)

Authors’ response: 

We thank the reviewer for the insightful suggestion. We have added the following description of the types of EVs and those which we will be isolating to the Introduction section at line 44-49 on page 3. 

“EVs can be divided into three main types, primarily based on their size and biological origin: (i) apoptotic bodies, which are greater than 800 nm diameter and are secreted by cells during apoptosis, (ii) microvesicles, which are large, membrane-bound vesicles (50–1,000 nm in diameter) that bud from the plasma membrane, and (iii) exosomes, which are 30–150 nm in diameter and of endocytic origin. In this study, EVs are defined as microvesicles and exosomes.”

Please move your list of antiboides located under "cell culture and antibodies" to the section titled "protein-based centrifugation and western blot analysis"

Authors’ response:

We have moved our list of antibodies to the “protein-based centrifugation and western blot analysis” section.

Please state the passage number used of HUVECs

Authors’ response:

We have added the passage number of HUVECs at line 115 on page 6.

“HUVECs up to passage 6 were used in this study.”

Please move the latter portion of lines 232-234 to the discussion section (speculation belongs in the discussion)

Authors’ response:

We have moved these lines to the discussion section (line 418-420 on page 19).

Please briefly summarize the findings found in reference 29 (lines 357-358 on page 16)

Authors’ response: 

We have added brief summary of the findings of the indicated reference (now on page 19, lines 411-414):

“In a previous study comparing the Invitrogen kit with differential centrifugation, the Invitrogen kit produced a higher yield of total protein and RNA than that produced by differential centrifugation due to the presence of non-EV proteins; the Invitrogen preparation also showed a broader size distribution [31].”

Please also include in the discussion that it is difficult to make decisions without: miRNA profiling, protein profiling, and more functional assays

Authors’ response:

We have added this suggestion to the discussion section (line 457-460 on page 21).

“…while our study showed that the different methods produced separated EV fractions with significantly different physical and functional properties, further studies that include miRNA profiling, protein profiling, and more functional assays of EVs isolated with each method are necessary.”

Reviewer #2: Summary

The goal of this research was to find the method that will best facilitate identifying EV biomarkers and thus identifying the method that is the most suitable for “follow-up studies involving EVs.” The authors isolated EVs by new and traditional approaches including differential ultracentrifugation, PEG precipitation with two new EV isolation technologies Exodisc and Exolute. For each isolation method then compared select DNA, RNA, and protein cargos as well as their ability to induce compliment activation. They found that each method distinct outcomes, isolating different amounts of particles with distinctive mean sizes, with different levels of select protein, RNA, and DNA cargos. They went on to show that the EV isolated from herpes-infected cells though the four methods induce different levels of compliment activation in naïve cells.

Major Concerns

As an EV biologist I can appreciate the findings in this manuscript. The Exodisc, and Exolute purification methods were news to me. It’s nice that they were able to directly compare these novel methods with two widely used purification methods. However, I was curious why they did not also compare immunoprecipitation as this is increasingly becoming the purification method of choice. The key finding of interest for me was that the Exodisc recovers way more EVs than other methods. Their results clearly show that the particles purified by the Exodisc contain traditional protein cargos of EVs and that these contain classical EV protein markers.

Authors’ response: 

We thank the Reviewer for their positive comments. We hope this study will help researchers working with EVs and are encouraged that the Reviewer found our Exodisc findings to be of interest.

I completely agree that immunoprecipitation is also useful for separating EVs. However, as the Reviewer indicated, our key finding was that Exodisc was the best way to obtain a larger quantity of EVs from a limited sample volume. For our purposes, immunoprecipitation was excluded because the result was expectied to produce a low yield. Furthermore, a previous study had already compared immunoprecipitation with other classical methods (Patel GK, Khan MA, Zubair H, Srivastava SK, Khushman M, Singh S, et al. Comparative analysis of exosome isolation methods using culture supernatant for optimum yield, purity and downstream applications. Sci Rep. 2019;9(1):5335).

While the authors conducted rigorous experiments that each showed extremely little variation between biological replicates. As it stands, they are only quantifying a few select RNA, DNA transcripts. It would be nice to have seen a broader characterization of the protein and RNA. In the background they describe their motivation is to identify a purification method that empowers RNAseq and LC-MS-MS analysis of EVs. It would have been nice ideal to have included some -omic data. 

Authors’ response:

We thank the Reviewer for the excellent observations and agree that -omic data would be ideal. Unfortunately, we faced technical and financial limitations. In terms of technical limitations, we did in fact conduct a proteomics study with EVs from KSHV-infected cells using the Exodisc method, but it was difficult to isolate a sufficient amount of EVs using the other methods. In terms of financial limitations, proteomics analysis is financially burdensome, and we had insufficient funding. We apologize that we could present only the results from our general analysis, but we believe that our results will still be helpful to other researchers in this field. 

Later in the discussion they establish their aim to determine “which method is most suitable for a specific research, specifically, we investigated which method would be most suitable for identifying biomarkers in EVs.” Later they change and say “It may be difficult for a single standard method to serve the purposes of all EV studies. We hypothesized that different methods are suitable for different EV study purposes.” And concluded with “In this study, physical and functional properties were analyzed and compared in EVs obtained using the different separation methods to demonstrate which methods would be suitable for follow-up studies involving EVs.” This is pretty vague, but even given this broad goal it’s not clear how these experiments are anything but the first step towards that. While the experiments are well done I feel that while these results are useful and necessary for the research group to develop their experimental methodology I don’t think that the novelty and impact of them is sufficient for publishing in PlosOne.

Authors’ response:

We thank the Reviewer for the helpful comments. According to the reviewer’s observation, we have revised the discussion section as follows:

“In conclusion, we provided novel data that showed how the recently developed ExoLutE and Exodisc methods compared to established methods. We demonstrated that the newly developed Exodisc method yielded the largest quantity of EVs; this indicated that Exodisc might be especially useful when searching for biomarkers. Selecting an appropriate separation method for a given purpose may be critical not only for producing useful results but also for reproducing experiments; our findings may help researchers determine which methods best suit their needs.”

Minor Concerns

The methods for EV purification reference other publications. It would be nice to have them at least described briefly. This seems especially important since the focus of the paper is on addressing the results of these methods.

Authors’ response:

We thank the Reviewer for the kind comments. 

We have added a further description of the detailed process for each method (line 127-150 on page 6~7).

The figures are difficult to follow without figure legends.

Authors’ response:

To address this issue, we have added a schematic summary to Figure 6 (Figure 6A).

For each experiment it would be nice to indicate the number of biological replicates.

Authors’ response: 

We have added the number of biological and technical replicates in the data set included in the Supporting Information file. 

Nanoparticle tracking analysis of EVs can be notoriously prone to artifacts and strongly dependent on setting conditions. Therefore, it would have been nice to provide a complimentary characterization of EV abundance, such as also show electron micrographs of EVs.

Authors’ response:

We agree that nanoparticle tracking analysis is dependent on the setting conditions. Therefore, we ran a nanoparticle tracking analysis with a constant setting condition, as described in the Materials and methods (maximum area: 1000, minimum area: 10, minimum brightness: 25, sensitivity: 75, shutter: 100, and temperature: 25 °C). With a constant setting condition, nanoparticle tracking analysis is a reliable tool for measuring the EV particle number. 

EM is a good tool for visualizing the EVs, but the field is too small to measure the particle number. In our previous study, EVs from HUVECs and KSHV-HUVECs were shown using EM (Jeon H, Yoo S-M, Choi HS, Mun JY, Kang H-G, Lee J, et al. Extracellular vesicles from KSHV-infected endothelial cells activate the complement system. Oncotarget; Vol 8, No 59. 2017). Therefore, we excluded EM results from this manuscript.

Line 163 “random RT primers were used for synthesis of 164 glyceraldehyde 3-phosphate dehydrogenase (GAPDH) and β-actin.” What do you mean by random? Primers of random sequence would not specifically amplify GAPDH or B-actin.

Authors’ response: 

Thank you for bringing our attention to this mistake. “Random RT primers” has been replaced with “random hexamers”. 

Also, the authors should make a case for why they chose to look at these transcripts.

Authors’ response:

Thank you for the suggestion. We have added a sentence at line 173-174 on page 8:

“To analyze the quality of mRNA, the housekeeping genes GAPDH and β-actin were used as representatives.”

They never state the number of particles they isolated with the different methods and the lo-res figures make it impossible to make out the exponential on the Y-axis.

Authors’ response: 

Please download the high-resolution image on the Plos One web page; it appears low-resolution only in PDF files.

There is little discussion on the significant differences they found in DNA and RNA cargo abundance. It would be nice to hear how finding differences in these nucleotide cargos furthers the overall goal of the research.

Authors’ response:

We thank the reviewer for the insightful suggestion. RNA was discussed in lines 402-405. We have added the following (lines 405-409): 

“The quantity of DNA in the EVs was too small to measure with a spectrophotometer, so qPCR was used for analysis. The qRT-PCR results shown in Fig. 4 demonstrated significant quantitative differences in genomic and mitochondrial DNA. EVs isolated with Exodisc had the highest quantity of DNA. However, relative to the EV particle number, the purity of the EVs isolated using the Invitrogen kit was higher.”

It would be nice to see Western blots also conducted on samples normalized for protein. As it is the Western blot results in figure 5B could be simply be due to the differential abundance of the starting samples.

Authors’ response: 

We thank the Reviewer for the insightful suggestion. We have uploaded western blots conducted on normalized protein samples as S1 Fig. and added a description at line 212-213 on page 10 and line 358~362 on page 17.

The authors do not make clear if their biological activity of inducing complementation is definitively associated specifically with EVs or there may also be other secreted factors that induce it. This knowledge is critical to make any kind of determination of EV bioactivity. Also it is curious that the ultracentrifuged EVs, with many less particles and lower EV protein makers nevertheless had similar activity to the much more abundant Exodisc sample. This would seem to show that the ultracentrifuged EVs was much purer in terms of activity. Maybe this could be due to a ceiling effect and which they could address with a dilution series.

Authors’ response: 

In our previous study (Jeon H, Yoo S-M, Choi HS, Mun JY, Kang H-G, Lee J, et al. Extracellular vesicles from KSHV-infected endothelial cells activate the complement system. Oncotarget; Vol 8, No 59. 2017), we showed that complement activation is specifically induced by EVs from KSHV-infected HUVECs. 

We agree that it was surprising that the ultracentrifuged EVs did not have better complement activation than the EVs in the Exodisc sample. Therefore, we described this issue in the discussion section at line 440-445. 

We were unable to determine which factor in the EVs stimulates the complement system. We suggested that differential centrifugation may be more effective for maintaining EV properties because of the higher purity.

Grammar issues throughout.

Authors’ response:

The manuscript has been edited by a professional English editing company. Please see the editing certificate.

---

## [Decision Letter · Decision Letter 1]

23 Jun 2020

Effects of different separation methods on the physical and functional properties of extracellular vesicles

PONE-D-20-09529R1

Dr. Lee,

We’re pleased to inform you that your manuscript has been judged scientifically suitable for publication and will be formally accepted for publication once it meets all outstanding technical requirements.

Kind regards,

Ramani Ramchandran

Academic Editor

PLOS ONE

Additional Editor Comments (optional):

Reviewers' comments:

Reviewer's Responses to Questions

**Comments to the Author**

1. If the authors have adequately addressed your comments raised in a previous round of review and you feel that this manuscript is now acceptable for publication, you may indicate that here to bypass the “Comments to the Author” section, enter your conflict of interest statement in the “Confidential to Editor” section, and submit your "Accept" recommendation.

Reviewer #1: All comments have been addressed

Reviewer #2: All comments have been addressed

2. Is the manuscript technically sound, and do the data support the conclusions?

Reviewer #1: Yes

Reviewer #2: Yes

3. Has the statistical analysis been performed appropriately and rigorously? 

Reviewer #1: Yes

Reviewer #2: Yes

4. Have the authors made all data underlying the findings in their manuscript fully available?

Reviewer #1: Yes

Reviewer #2: Yes

5. Is the manuscript presented in an intelligible fashion and written in standard English?

Reviewer #1: Yes

Reviewer #2: Yes

6. Review Comments to the Author

Reviewer #1: All points raised regarding the manuscript were sufficiently addressed. The manuscript is now technically sound, and statistical analysis have been included in a supplemental file. Additionally, minor corrections raised regarding lack of citations and definition of certain terms have been corrected.

Reviewer #2: The authors addressed my technical concerns and refocused the article to better reflect what their experiments were capable of showing. They were direct in identifying the limitations of their study. Therefore if the editor feels the results have sufficient biological interest for publication then I will not dispute.

7. PLOS authors have the option to publish the peer review history of their article (what does this mean?). If published, this will include your full peer review and any attached files.

Reviewer #1: No

Reviewer #2: Yes: Joshua C. Russell

---

## [Editor Report · Acceptance letter]

25 Jun 2020

PONE-D-20-09529R1 

Effects of different separation methods on the physical and functional properties of extracellular vesicles 

Dear Dr. Lee:

I'm pleased to inform you that your manuscript has been deemed suitable for publication in PLOS ONE. Congratulations! Your manuscript is now with our production department. 

Kind regards, 

on behalf of

Dr. Ramani Ramchandran 

Academic Editor

PLOS ONE